# Morphoanatomical Changes in *Eucalyptus grandis* Leaves Associated with Resistance to *Austropuccinia psidii* in Plants of Two Ages

**DOI:** 10.3390/plants12020353

**Published:** 2023-01-12

**Authors:** Edson Luiz Furtado, André Costa da Silva, Érica Araújo Rodrigues Silva, Roberto Antônio Rodella, Marcus Alvarenga Soares, José Eduardo Serrão, Cristiane de Pieri, José Cola Zanuncio

**Affiliations:** 1Departamento de Proteção Vegetal, Faculdade de Ciências Agronômicas, Universidade Estadual Paulista, Botucatu 18610-034, São Paulo, Brazil; 2Departamento de Fitossanidade, Universidade Federal do Rio Grande do Sul, Porto Alegre 91540-000, Rio Grande do Sul, Brazil; 3Instituto de Pesquisas e Estudos Florestais (IPEF), Piracicaba 18618-000, São Paulo, Brazil; 4Departamento de Botânica, Instituto de Biociências de Botucatu, Universidade Estadual Paulista, Botucatu 18618-000, São Paulo, Brazil; 5Programa de Pós-Graduação em Produção Vegetal, Universidade Federal dos Vales do Jequitinhonha e Mucuri, Diamantina 39100-000, Minas Gerais, Brazil; 6Departamento de Biologia Geral, Universidade Federal de Viçosa, Viçosa 36570-900, Minas Gerais, Brazil; 7Departamento de Entomologia/BIOAGRO, Universidade Federal de Viçosa, Viçosa 36570-900, Minas Gerais, Brazil

**Keywords:** anatomy, eucalypts, leaf maturation scale, leaf stages, Myrtaceae rust, morphological markers, ontogenic resistance

## Abstract

The fungus *Austropuccinia psidii* infects young tissues of *Eucalyptus* plants until they are two years old in the nursery and field, causing Myrtaceae rust. The characteristics making older eucalypt leaves resistant to *A. psidii* and the reason for the low levels of this pathogen in older plants need evaluations. The aim of this study was to evaluate the morphological differences between *Eucalyptus grandis* leaves of different growth stages and two plant ages to propose a visual phenological scale to classify *E. grandis* leaves according to their maturation stages and to evaluate the time of leaf maturation for young and adult plants. A scale, based on a morphological differentiation for *E. grandis* leaves, was made. The color, shape and size distinguished the leaves of the first five leaf pairs. Anatomical analysis showed a higher percentage of reinforced tissue, such as sclerenchyma-like tissue and collenchyma, greater leaf blade thickness, absence of lower palisade parenchyma in the mature leaves and a higher number of cavities with essential oils than in younger ones. Changes in anatomical characteristics that could reduce the susceptibility of older *E. grandis* leaves to *A. psidii* coincide with the time of developing leaf resistance. Reduced infection of this pathogen in older plants appears to be associated with a more rapid maturation of their leaf tissues.

## 1. Introduction

*Eucalyptus* plantations in Brazil are among the most productive in the world, with approximately 40 m^3^ ha^−1^ year^−1^ [1,2]. Adequate environmental conditions [3] and physiological adaptations [4] favor this productivity. Additionally, rapid growth and high yield of *Eucalyptus* clones can improve pulpwood production [5]. *Eucalyptus grandis* (W. Hill) Maiden is one of the most commonly used for this purpose [6]. This species has an excellent genetic basis [4,7,8], but most of its genotypes are susceptible to *Austropuccinia psidii* (G. Winter) Beenken [4,5,9]. This fungus, with a wide geographical distribution, is present in four of the five continents threatening Myrtaceae biodiversity, mainly in the *Eucalyptus* forestry industry [10,11].

*Austropuccinia psidii*, the etiologic agent of Myrtaceae rust, occurs mainly in *Eucalyptus* nurseries and on young plants in the field. This fungus is found on plant parts younger than two years [10], such as leaf primordia, young leaves and petiole. It causes necrosis and deformation of tissues with yellowish pustules in highly susceptible hosts [12,13,14]. *Austropuccinia psidii* generally penetrates directly through the cuticle and epidermis by *appressorium* [1,15,16]. However, the entrance, establishment and reproduction of this fungus gradually decrease with the *Eucalyptus* leaf age [15,17]. *Austropuccinia psidii* does not infect the leaves of susceptible eucalypt plants from the fifth leaf stage onward [15,16].

The effect of plant-growth stages on pathogen resistance is age-related resistance, developmental resistance or ontogenic resistance and has been reported in *Myrtaceae* [18] and other plants [19]. Host tissue susceptibility may increase or decrease over time according to the disease [20]. Resistance at different host development stages varies with plant age or tissue maturity and may be specific or broad-spectrum, driven by diverse mechanisms, depending on plant–pathogen interactions [19]. The influence of the phenological stage of *Eucalyptus* leaves on *A. psidii* resistance and the factors making older leaves resistant need to be better understood. This represents a key step to developing a viable method to manage eucalypt rust efficiently [1]. The effects of *Eucalyptus* leaf age on the production of antimicrobial compounds and on *A. psidii* pre-infection success have been studied [15,21,22,23], but the relation between leaf anatomy and resistance to rust has not. This information may improve our understanding of the phenological leaf age and resistance to eucalyptus rust.

The morphological characteristics of *E. grandis* leaves at different growth stages were studied. A visual scale to classify the eucalypt leaves relative to their stage of maturation and the time for leaf maturation of young and adult plants in the field using a scale was proposed. The hypothesis was that the age-related resistance of *E. grandis* to *A. psidii* includes morphological defense mechanisms.

## 2. Materials and Methods

### 2.1. Morphological Characteristics

A visual phenological scale for *E. grandis* leaves of young and adult plants at six and 20 months old, respectively, is proposed. The evaluation was conducted in Santa Branca (23°24’ S and 45°51’ W), São Paulo, Brazil, with quartizarenic soil in the area. Cultural treatment and plant fertilization were applied as recommended by the forest company (calcium nitrate 1.0 g L^−1^; MAP 0.25 g L^−1^; potassium nitrate 0.75 g L^−1^; magnesium sulfate 0.7 g L^−1^; sulfate ammonium 0.33 g L^−1^ and micronutrient solution 0.15 mL L^−1^ every six months until the plants were two years old). Thirty branches of young and adult plants were collected at the top, median and lower plant heights from three *E. grandis* clones in the field (TC31, VR3748 and C041A) at “Votorantim Celulose e Papel (VCP)” commercial plantations, with 6 to 10 leaf pairs per branch, at different development stages per clone and age (Appendix A). These leaves were detached from the branches with their petiole, separated into pairs according to their position from the tip to the base of the branches and visually rated. The apical bud and the leaf pair nearby (closed) were numbered as growth leaf stage “zero” and not considered in this study. The subsequent opened leaf pairs were numbered growth stages 1, 2, 3, 4 and 5, grouped by similarity per class. The first pair considered was the very red one; the second pair with leaves larger than the previous and also red; average ones with color tending from slightly red to green; green and older leaves had more rigid appearance and wider than previous stages; and those with larger size than the others, with an intense green color, respectively. The subsequent leaves had no significant change in color or size compared to the fifth pair and, thus, they were not considered. The leaf length, width, area and shape; the shape of the leaf apex; the petiole diameter; and the leaf color were evaluated. The length, width and leaf area were measured using a scanner; the petiole diameter with a digital caliper rule (Digimess Resolution of 0.01 mm); the shape of the leaf apex was determined [24]; and the leaf color evaluation was based on the Exotic Encyclopedia Color Chart [25].

### 2.2. Anatomical Characteristics

The leaf growth stages were defined and used in scale proposed with three leaves per developmental class, from 6- and 20-month-old plants classified according to their anatomical characteristics. The leaves per stage were cut in the third middle region of the midrib and the adjacent internervural area. The leaf cuts were fixed in a solution of formaldehyde, acetic acid and 70% ethanol (5:5:90 *v*/*v*) for 48 h (Johansen, 1940) and stored in 70% ethanol. The samples were dehydrated in a graded ethanol series and embedded in glycol methacrylate resin [26] and transverse sections of approximately 8 µm were sectioned with rotary microtome (RM 2145, Leica). Fifteen fragments were obtained per sample. Sections were stained with 0.05% toluidine blue in sodium phosphate buffer [27] and mounted in synthetic resin (Permount-Fisher, Suwanee, GA 30024, USA). The leaf structures were analyzed in triplicates, drawn with the aid of a projection microscope Axioskop 40 coupled to an optical microscope Zeiss (Axio Vision Camera and projection Axio Viewer), and the measurements of the structural characters carried out using a table scanner coupled to area meter. The following anatomical characteristics of the interveinal region of the leaf blade were quantified: leaf thickness, cuticle, epidermis and palisade parenchyma of adaxial and abaxial faces. The percentage of palisade parenchyma and spongy parenchyma in relation to leaf thickness, area and number of oil cavities per mm^2^ of leaf surface was calculated. The anatomical sections were embedded in historesin and stained with Sudan IV before measuring the thickness of the adaxial and abaxial cuticle.

### 2.3. Leaf Differentiation over Time

The periods for changes between growth stages of *E. grandis* C041H leaves were assessed using the scale proposed. The evaluation was conducted in the same soil type, cultural treatment and fertilization, where branches were collected during the proposed scale preparation. Ten 6-month-old plants and ten adult 20-month-old plants were evaluated. Ten lateral branches were marked in the middle third of each tree. The leaves of each stage were marked with a different color. The plants were evaluated daily according to the proposed scale (Figure 1) and the time taken for the leaves to reach the next stage was evaluated over 60 days. The average temperature and rainfall during the experiment were 23.5 °C and 21.2 mm, respectively.

### 2.4. Statistical Analysis

The variance in the results of quantitative characteristics of the leaf anatomical analysis was calculated and the means compared by Tukey test at 5% probability and submitted to multivariate statistical tests of cluster and principal component analysis (PCA) [28]. Principal component analysis allowed to reduce the number of variables with the least possible information losses. The Y1 and Y2 axes represented the directions with maximum variability (which allow for a simpler interpretation of the structure of covariance matrix) with methodology proposed by Sneath and Sokal [28]. The correlation values between the 13 anatomical characters and the two main components (Y1 and Y2) were studied. Cluster analysis of quantitative anatomical leaf characteristics was performed with the Euclidean distance average between the five leaf growth stages and graphically demonstrated by dendograms. Data for leaf differentiation over time were analyzed with two factors (age and two rankings per leaf development: 1–3 and 3–5) and the variable was the number of days. The variance and treatment means were compared by Tukey test at 5% probability. The normality and homogeneity were examined using the residues without deviations based on the presuppositions of the analysis. The statistical analyses were performed using R (R Development Core Team) software.

## 3. Results 

### 3.1. Morphological Characteristics

The qualitative and quantitative characteristics differentiated the five main phenological *E. grandis* leaf stages (Table 1, Appendix A, Appendix A and Appendix A) and the photographic visual scale accurately identified them (Figure 1). The qualitative (color and shape) and quantitative characteristics of the leaf blade (diameter of the petiole, leaf area, length and width) per development stage were similar between 6- and 20-month-old plants. The apex shape (sharp) and the leaf blade color (chocolate) differentiated the first leaf stage from the others (Table 1 and Figure 1). Staining Ivy Green (number 70 of the color letters) discriminated the fifth leaf development stage. The three intermediate stages (2, 3 and 4) were similar with an ovoid shape, sharped apex and olive-green color (84), making it necessary to individually measuring them (Table 1 and Figure 1). The leaf area, length and width distinguished leaf stages 2, 3 and 4 with values of 2.3–3.8, 7.2–8.9 and 19.2–27.5 cm^2^ of leaf area, 3.5–4.3, 5.0–6.2 and 8.3–9.5 cm long, 1.2–1.5, 2.2–2.5 and 3.7–4.7 cm wide, respectively.

### 3.2. Anatomical Characteristics

The increase or decrease in the percentage of tissues over the five leaf-growth stages were similar in the leaf midribs of the 6- and 20-month-old plants, but it varied with the leaf development stage (Table 2 and Figure 2). The percentages of vascular bundles and collenchyma increased from stages 1 to 4 and with lower values at stage 5. Leaves of 6-month-old plants had no collenchyma during stage 1, which was found from the beginning of leaf development in 20-month-old plants and with an expressive increase in leaf stages 3 and 4. The parenchyma decreased in leaf stages 1 to 4 and increased in stage-5 leaves. The sclerenchyma-like tissue was found only in stage-5 leaves with significant thickening near the epidermis. The total area of the midrib increased during all leaf development stages. The total epidermal tissue in the internervural leaf region of young plants decreased in the upper and lower surfaces according to leaf development. In contrast, those from adult plants showed no apparent tendency for variation in the percentage of these tissues over time (Table 3 and Figure 2). The upper-leaf palisade parenchyma increased from stages 1 to 3 in young and adult plants and decreased in the following ones. The lower palisade parenchyma of younger plants increased up to stage 2, was stable during stage 3 and decreased until it disappeared at stage 5. The percentage of this tissue decreased throughout leaf development and disappeared in stages 4 and 5 in adult plants. The spongy parenchyma decreased until stage 3, but with a significant increase from then on for the leaves of younger plants. The spongy parenchyma of adult plants decreased until stage 2, doubled in percentage from stages 3 to 4 and maintained a high percentage during stage 5. The spongy parenchyma increased and, consequently, the upper palisade decreased, especially at stage 3. Leaf thickness and the total area of oil cavities of young and adult leaf plants increased from stages 1 to 5, especially during stages 4 and 5 (Table 3 and Figure 2). The first leaf stage had no oils in the cavities, but their numbers increased with leaf development (Figure 2). A single cavity with oil was found in the leaves of stages 2 and 3 and all cavities in stages 3 and 4 had essential oils. The total area of the leaf internervural regions increased with plant development.

The five leaf development stages of young and adult *E. grandis* plants differed, based on the dendrogram (Figure 3) and graphic dispersion (Figure 4), resulting from 13 quantitative anatomical descriptors of the midrib and internervural region (Table 4). Cluster analysis presented four groups, at 0.15 to 0.2 on the Euclidean distance scale (Figure 3). Young eucalypt plants had the first, second, third and fourth groups formed by the first; second and third; fourth; and fifth development leaf stages, respectively (Figure 3A). The second and third leaf stages are anatomically closer to stage-1 leaves than to those of 4 and 5. The first, second, third and fourth groups for mature eucalypt plants were constituted by the first and the second leaves; third; fourth; and fifth leaf stages, respectively (Figure 3B). The first and second leaf stages were more anatomically similar than the others were. The third group, formed by leaf stage 3, was anatomically similar to stages 1 and 2. The convergence of other groups occurred at a very advanced scale level, demonstrating differences in leaf anatomy between them. Leaf stages 1, 2 and 3 are the most susceptible to rust and are anatomically similar, whereas those of leaf stages 4 and 5 are more resistant to the pathogen and anatomically similar. The correlation values between the 13 anatomical characters and the two main components (Y1 and Y2) of the young and adult plants, respectively, grouped the leaf stages in an identical manner to the cluster analysis (Figure 4). The correlation values were close to 1 with high discriminatory power for the leaf stages (Table 4). The major contribution of anatomical characters related to Y1 values accounted for 86.4 and 80.5 of the accumulated information for young and adult plants, respectively. Reinforcement tissues, such as sclerenchyma-like tissue, collenchyma and vascular bundles, in addition to leaf thickness and cavities containing essential oils, were the major influences on the Y1 axis, consistent with the hypothesis that the concentration of reinforcement tissues and cavities with essential oils are important for leaf resistance to rust.

### 3.3. Period of Leaf Differentiation

The period of leaf stage changes from 1 to 3 and 1 to 5 on young plants was longer than on adult ones (Table 5). The changes in the leaf stages from 1 to 3 and 1 to 5 were 34, 53, 24 and 43 days for young and adult plants, respectively. The initial stages during maturation were longer in leaves of young plants (1, 2 and 3).

## 4. Discussion

The proposed visual scale (Figure 1) will allow for the classification of the eucalypts leaf of interest according to its development stage using a simple comparison or by its length or width using graphic interpolation (Appendix A: Appendix A and Appendix A).

The differentiation of the first five growth *E. grandis* leaf stages is important to quantify eucalypt rust disease, the resistance of different eucalypt genotypes and to select leaves for deposition of inoculums in resistance tests [15,16]. This made it possible to evaluate eucalypt rust severity, since the leaves are similar from this stage onward and no infection by *A. psidii* was found after stage 5. The classification into five growth stages based on leaf ontogeny and an image scale for leaf differentiation is widely used for rubber tree leaves [29] to evaluate a *Pseudocercospora ulei*–rubber tree pathosystem [30,31]. The quantification of rust on eucalypt leaves and their resistance to this disease must be carried out using the first to third leaf pairs, since the disease decreases after the third leaf stage onward [15,22]. The apex shape (acute) and leaf blade color (chocolate) easily distinguish the first leaf pair; however, the second, third and fourth leaf pairs are similar, making the measurement of their length, width or even leaf area necessary to distinguish them.

The reinforced tissues, such as sclerenchyma, collenchyma, vascular bundles and leaf thickness, with the increasing age of the *E. grandis* leaf stages, based on anatomical analysis, show the importance of these tissues in the resistance to pathogens [32,33,34,35]. The anatomical analysis of the plant can evidence the pathogenesis and structural differences that make them generally or partially resistant to pathogens [36]. *Austropuccinia psidii* generally directly penetrates eucalypt leaves, through the cuticle and epidermis, by forming *appressorium* [15,16]. The damage by this fungus, mainly in the first to third leaf pairs, and the decreasing number and size of lesions from the fourth pair tending to zero on subsequent leaf stages make older leaves resistant, even in susceptible clones [22]. The entrance, colonization and reproduction of *A. psidii* decrease dramatically from the third leaf stage, dropping to zero from the fifth eucalypt leaf stage [15,22] (Appendix A), which may be due to the increasing leaf tissue reinforcement as eucalypts with mature leaves. The reinforced leaf tissues act as a physical barrier preventing pathogen penetration and growth [32,33,35]. The anatomical similarities between leaf stages 1, 2 and 3 (more susceptible to rust) than leaf stages 4 and 5 (more resistant to rust) have been reported [22]. The cluster and main component analyses reveal the relationship between the quantities of leaf reinforcement tissues and increasing leaf age. The decrease until the total absence of the lower palisade parenchyma in stages 3 to 4 and 4 to 5, respectively, in 6-month-old plants and the decrease in stages 1 to 2 and 2 to 3 until complete absence in stages 4 and 5 in the leaves of 20-month-old plants are important. Isobilateral mesophyll at the early stage of leaf growth is expected in Myrtaceae species, such as *Calistemon* and *Eucalyptus*; however, they become dorsal–ventral in mature stages [37]. This phenomenon should affect the colonization by the pathogen due to the lack of this tissue.

The deposition of lignin with the transformation of collenchyma to sclerenchyma during leaf stage 5 may be important to prevent the penetration of *A. psidii* in the eucalypt leaves due to the multiple layers surrounding the leaf interior. This forms a thicker zone with lignified and stronger tissues [33,38], protecting the parenchyma and vascular bundles from fungal invasion [33,34]. The thicker sclerenchyma cells observed in this study near the epidermis formed a stronger barrier against *A. psidii* penetration, as reported for *Magnaporthe oryzae* penetration in rice plants [33,39]. The sclerenchyma cells are important for the immune response to pathogen invasion [33] and greater thickness and higher number of cavities with essential oils are also linked to the resistance to *A. psidii*. This increase, in addition to the quantity of wax on older leaves and the chemical composition of cuticle waxes, may interfere with *A. psidii* infection on the older eucalypt leaves [15,16,40]. The antimicrobial compounds, including chitinase and peroxidase enzymes, polyphenols and tannins, monoterpenes, steroids, monounsaturated hydrocarbons, limonene and essential oils in older eucalypt leaves [21,22,23], reduced the germination, penetration and colonization processes in this fungus [15,16]. The increase in the number of cavities with essential oils, mainly at the 4th and 5th leaf stages, is important because these oils have a complex of volatile organic compounds [41] toxic to plant pathogens [42,43,44]. The limonene in the essential oil is toxic to *A. psidii* and its quantity increases with eucalypt leaf age [22]. The formation of reinforced tissues and the increase in the number of oil cavities should be related to the resistance of the older eucalypt leaves to *A. psidii* infection. This information improves our understanding of the phenological leaf age and resistance to *A. psidii* infection.

The leaf scale proposed made it possible to evaluate the periods of leaf-stage change in 6-month-old and 20-month-old *E. grandis* plants. Younger plants may be more vulnerable to rust because they remain longer during the initial stages (1, 2 and 3) and with a longer maturation period. The transition from stages 1 to 3 and 1 to 5 was 10 days faster in older than in younger plants, which may explain no occurrence of this disease in the former in the field due to leaf escape during periods of high susceptibility. This phenomenon may reduce the occurrence of the disease due to a shorter exposure period for leaves more susceptible to rust and this could reduce the epidemic period [1].

The higher resistance of adult plants (20-month-old *E. grandis*) and older leaves (above the fourth leaf stage) of *Eucalyptus* spp. to rust is similar to that found in rubber- *Pseudocercospora ulei* pathosystems (leaves up to or older than 21 days are susceptible and resistant, respectively) and the higher occurrence of this fungus on young rubber trees is due to the new shoots sprouting every month [30]. The lower intensity of *A. psidii* in older plants may be due to a shortened period of high susceptibility associated with more rapid leaf maturation, similar to that reported for the *Micosphaerella* sp. and *Teratosphaeria* sp. on *Eucalyptus* plants [45]. The transition from young to adult leaves of the hybrid genotypes of *E. globulus* × *E. urophylla*, and *E. urophylla* × *E. grandis* is faster reducing the susceptibility to the *Mycosphaerella* spot [45]. Rare pathogens infect the host organs at all ages and stages and growth stages susceptible to infection are present in most of them [42]. The development of genetic materials with a faster leaf maturation rate may decrease the occurrence of *A. psidii,* both in the nursery and in the field, and is an important tool for breeders. This work provides additional information about the mechanisms of resistance of eucalypt plants to *A. psidii*.

## 5. Conclusions

This is the first report demonstrating the natural physical resistance mechanisms of older eucalypt leaves to Myrtaceae rust, based on morphological and anatomical characteristics, which may explain the lower occurrence of this disease in older eucalypt plants.

The occurrence of reinforced tissues, including sclerenchyma-like tissue and collenchyma, greater leaf blade thickness, vascular bundles and the high number of cavities containing essential oils, may contribute to older *E. grandis* leaves being more resistant to rust.

The absence of lower palisade parenchyma from the fourth leaf stage of young plants and those of third leaf stage in adult plants could be important to reduce tissue invasion by *A. psidii*.

The lower occurrence of the fungus *A. psidii* in older plants is probably due to the escape of its leaves due to the rapid maturation period (10 days) compared to younger plants under two years old.

## Figures and Tables

**Figure 1 plants-12-00353-f001:**
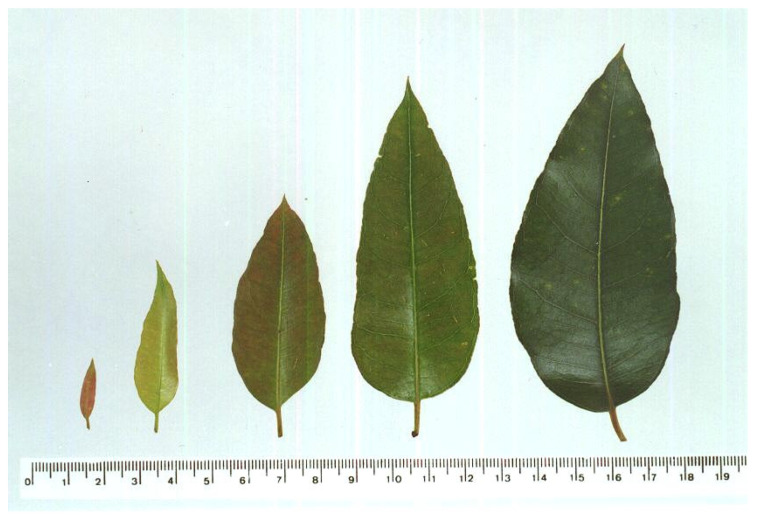
Visual scale of *Eucalyptus grandis* leaves at the five representative development stages from 6- and 20-month-old plants based on the shape and color of the leaf blade. A scale (cm) is below and the leaves were placed in order according to their position on the branch.

**Figure 2 plants-12-00353-f002:**
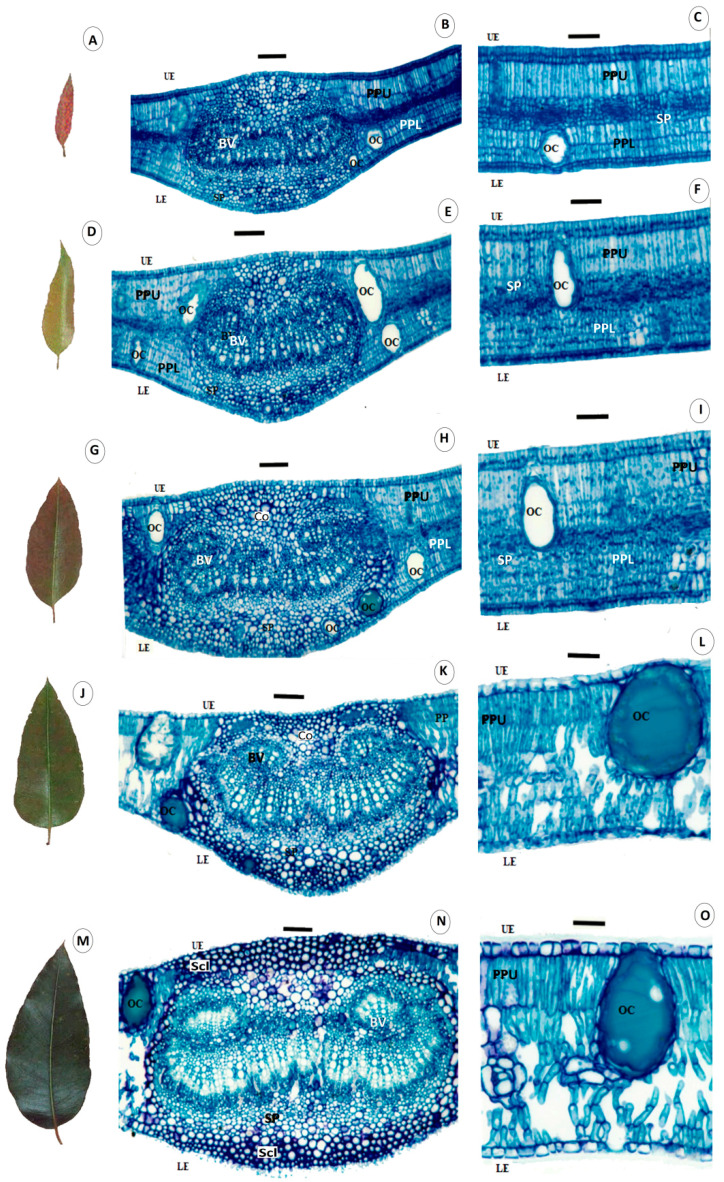
Light micrographs of the first (**A**–**C**), second (**D**–**F**), third (**G**–**I**), fourth (**J**–**L**) and fifth (**M**–**O**) leaf-growth stages of *Eucalyptus grandis*. Transverse sections of the midrib (**B**,**E**,**H**,**K**,**N**) and the internervural region (**C**,**F**,**I**,**L**,**O**). UE = upper epidermis tissue, LE = lower epidermis, PPU = palisade parenchyma upper, PPL = palisade parenchyma lower, SP = spongy parenchyma, Co = collenchyma, Scl = sclerenchyma-like tissue, OC = area of oil cavities, and BV = vascular bundles. Scale bars: 92 (**B**,**E**,**H**,**K**,**N**) and 46 µm (**C**,**F**,**I**,**L**,**O**).

**Figure 3 plants-12-00353-f003:**
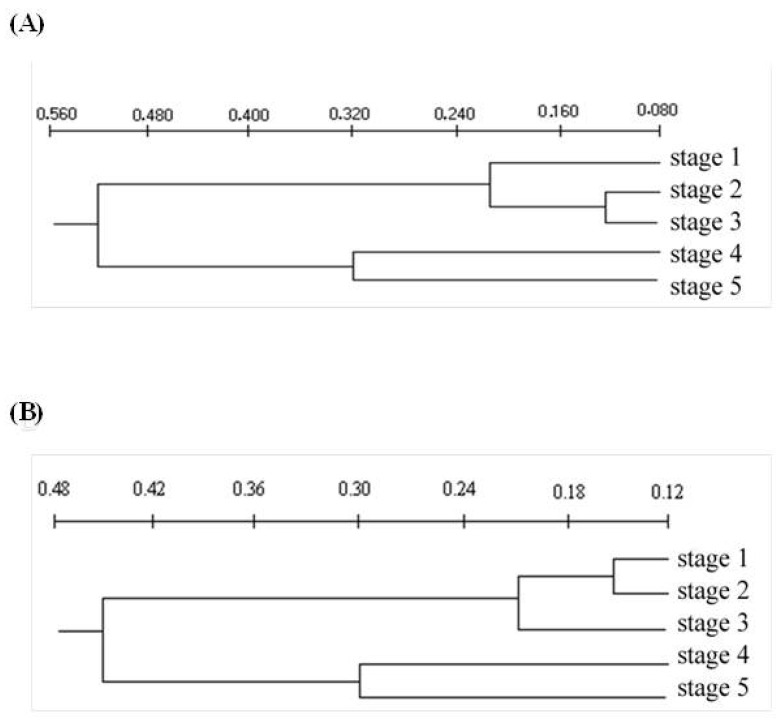
Cluster analysis of the quantitative anatomical variables for the five leaf development stages of *Eucalyptus grandis*. (**A**) Young (6 months old) and (**B**) mature (20 months old) plants.

**Figure 4 plants-12-00353-f004:**
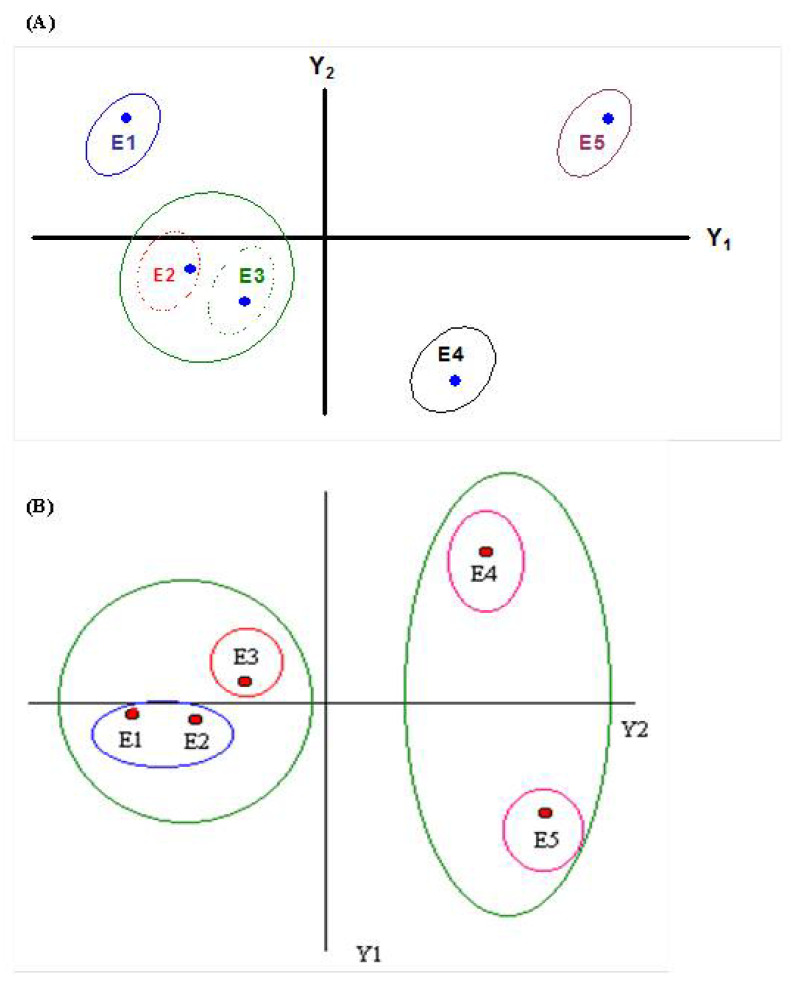
Dispersion graph for the five development stages of *Eucalyptus grandis* leaves using the two principal components (Y1 and Y2) for all the 13 quantitative anatomical descriptors of the midrib and internervural region. Dispersion graph of 6-month-old (**A**) and 20-month-old (**B**) plants.

**Table 1 plants-12-00353-t001:** Differentiation of leaf stages based on quantitative (length and width), area, diameter of the petiole and qualitative (shape and color) factors for the five phenological development stages of *Eucalyptus grandis* leaf development.

	Leaf Growth Stage				
Characteristic	1°	2°	3°	4°	5°
Leaf blade length (cm)	2.0(1.7–2.4) * ^a^ ***	3.8(3.5–4.3) ^b^	5.7(5.0–6.2) ^c^	9.0(8.3–9.5) ^d^	10.2(10.1–0.3) ^e^
Leaf blade width (cm)	0.6(0.5–0.7) * ^a^ ***	1.3(1.2–1.5) ^b^	2.3(2.2–2.5) ^c^	4.2(3.7–4.7) ^d^	5.1(4.8–5.7) ^e^
Leaf area (cm^2^)	0.5(0.3–0.6) * ^a^ ***	2.9(2.3–3.8) ^b^	8.0(7.2–8.9) ^c^	23.5(19.2–27.5) ^d^	31.8(30.4–33.2) ^e^
Diameter of petiole (cm)	0.6(0.5–0.7) * ^a^ ***	0.9(0.8–1.0) ^b^	0.96(0.9–1.0) ^c^	1.31(1.2–1.38) ^c^	1.5(1.4–1.6) ^d^
Leaf shape ^1^	Acute 73% **	Obtuse 73%	Obtuse 73%	Obtuse 60%	Obtuse 73%
Leaf color ^2^	Ch (7) 71% **	OG (84) 53%	OG (84) 51%	OG (84) 84%	IG (70) 98%

* Minimum and maximum values of the quantitative characters; ** Frequency of descriptive characters. *** Means followed by the same letter per line do not differ by Tukey test at 5% probability. Leaf blade (SDM = 0.62; GM = 6.16 and Deviation = 0.33); Leaf blade width (SDM = 0.37; GM = 2.72 and Deviation = 0.19); Leaf area (SDM = 8.83; GM = 14.02 and Deviation = 4.67) Petiole diameter (SDM = 0.14; GM = 1.02 and Deviation = 0.08) ^1^ Leaf classification according to Hickey (1973). ^2^ Color Chart Encyclopedia Exotic, “Horticultural Color Guide” (GRAF, 1970). Ch—Chocolate, OG—Olive Green, and IG—Ivy Green.

**Table 2 plants-12-00353-t002:** Percentage of tissues in the upper and lower epidermis, vascular bundles, collenchyma, sclerenchyma-like tissue, parenchyma and total area of the midrib for the five growth stages of *Eucalyptus grandis* leaf development stages of 6- and 20-month-old plants.

Midrib-Percentage	Six Months					Twenty Months				
Of Tissues (%)	1°	2°	3°	4°	5°	1°	2°	3°	4°	5°
Upper epidermis	5.5	5.0	4.0	3.0	2.2	4.9	3.8	3.0	2.6	1.9
Lower epidermis	4.1	3.8	3.0	2.4	2.1	4.1	3.2	2.6	2.4	1.6
Vascular bundles	28.2	33.8	35.6	39.9	36.2	31.2	40.6	43.7	49.9	47.5
Collenchyma	0.0	9.1	9.2	16.2	12.8	6.4	7.0	7.4	12.0	7.9
Sclerenchyma-like tissue	0.0	0.0	0.0	0.0	2.3	0.0	0.0	0.0	0.0	2.3
Parenchyma	62.1	48.3	48.2	38.5	40.5	52.8	46.1	43.4	34.3	38.1
Total area (×10^3^ µm^2^)	207.7	274.3	413.1	479.9	756.0	220.8	287.2	382.5	505.3	881.9

**Table 3 plants-12-00353-t003:** Percentage of tissues in the upper epidermis, upper palisade parenchyma, spongy parenchyma, lower palisade parenchyma, lower epidermis, leaf thickness, area of oil cavities and total area in internerval region for the five growth stages of *Eucalyptus grandis* leaf development stages of 6- and 20-month-old plants.

Internerval Region	Six Months					Twenty Months				
	Leaf Stage									
Percentage of Tissues (%)	1°	2°	3°	4°	5°	1°	2°	3°	4°	5°
Upper epidermis	11.6	11.0	10.3	7.3	5.0	8.6	9.8	6.5	5.8	6.8
Upper palisade parenchyma	28.2	29.5	34.4	27.2	20.5	24.7	28.8	34.3	24.4	22.6
Spongy parenchyma	26.8	23.4	20.3	51.6	69.7	28.0	26.8	30.3	65.0	65.2
Lower palisade parenchyma	25.9	28.7	28.5	7.4	0.0	30.4	28.6	19.0	0.0	0.0
Lower epidermis	7.4	7.3	6.3	6.6	4.8	8.5	6.1	9.9	4.4	5.5
Leaf thickness (µm)	179.5	187.1	227.1	232.4	297.1	170.0	207.0	215.0	243.0	248.0
Area of oil cavities (×10^3^ µm^2^)	1.7	2.1	2.9	4.5	8.0	2.1	2.2	2.4	11.0	12.9
Total area (×10^3^ µm^2^)	95.0	97.0	115.6	122.1	153.0	83.9	101.4	118.7	122.4	131.6

**Table 4 plants-12-00353-t004:** Correlation coefficients between 13 anatomical characteristics of the midrib (Mi) and internerval region (IR) for the five development stages of *Eucalyptus grandis* of 6- and 20-month-old plants and their two main components (Y1 and Y2).

Anatomical Characteristics	Six Months				Twenty Months			
	Y1	Or. ^z^	Y2	Or.	Y1	Or.	Y2	Or.
Upper epidermis (Mi)	−0.9496	5	0.2338	7	−0.9210	4	−0.1511	9
Lower epidermis (Mi)	−0.9148	6	0.3071	6	−0.7052	9	−0.5045	3
Vascular bundles (Mi)	0.6240	13	−0.7686	1	0.8386	7	0.3474	6
Collenchyma (Mi)	0.7045	11	−0.6859	3	0.6611	12	0.6805	2
Sclerenchyma-like tissue (Mi)	0.8494	9	0.4774	4	0.6939	11	−0.7090	1
Parenchyma (Mi)	−0.6537	12	0.7302	2	−0.8941	6	−0.3756	5
Upper epidermis (IR)	−0.9962	1	0.0712	12	−0.7053	8	−0.4157	4
Palisade parenchyma upper (IR)	−0.8153	10	−0.4040	5	−0.5466	13	0.3013	7
Spongy parenchyma (IR)	0.9722	3	0.0792	11	0.9812	2	0.0835	12
Palisade parenchyma lower (IR)	−0.9649	4	0.0196	13	−0.9823	1	−0.1490	10
Lower epidermis (IR)	−0.8978	7	−0.1351	10	−0.7044	10	−0.1031	11
Leaf thickness (IR)	0.8848	8	0.1840	8	0.9147	5	0.1537	8
Area of oil cavities (IR)	0.9892	2	0.1354	9	0.9760	3	0.0761	13
Retained information (%)	86.44		13.56		80.50		19.50	
Accumulated information	86.44		100.00		80.50		100.00	

^Z^ Or.—Ordination of anatomical characteristics as the discriminatory power.

**Table 5 plants-12-00353-t005:** Average period (days) of leaf differentiation of growth stages 1 to 3 and 1 to 5 of *Eucalyptus grandis* plants of 6 and 20 months old.

Plant Age (Months)	Leaf Stage of Change (Days) *	
	1–3	1–5
6	34 ^aB^	53 ^aA^
20	24 ^bB^	43 ^bA^

Means followed by the same lowercase letter per column or capital letter per line do not differ by the Tuckey test at 5%. * Evaluation of development stages changes according to scale proposed (SMD = 0.3004; GM = 6.0889; deviation = 0.5112; data transformed in (x + 0.5)^1/2^).

## Data Availability

The data presented in this study are available on request from the corresponding author. The data are not publicly available because many of them are qualitative and most of the quantitative analyses were conducted by computerized images of plant tissue anatomy.

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
