# Peer review of "Morphoanatomical Changes in Eucalyptus grandis Leaves Associated with Resistance to Austropuccinia psidii in Plants of Two Ages"

_plants, 2023, doi:10.3390/plants12020353_

Round 1
Reviewer 1 Report
General comments:
This paper presents some interesting anatomical information that could be relevant to the way susceptibility to A. psidii infection decreases with increasing age of leaves. The study concerns the genetically susceptible combination of E. grandis being infected by A. psidii. In this host, as in all susceptible Myrtaceae hosts, infection by A. psidii occurs only in the top few leaf pairs near the growing apex of stems. Leaves lower than stem position 4-5 are always resistant.
Title: The current title does not adequately reflect what the paper is about. This is explained in the comments below, but I suggest the title should be changed to something like:
Morphology changes in Eucalyptus grandis leaves associated with resistance to Austropuccinia psidii
While this is an interesting anatomical study, the paper is unfortunately not publishable in its current form because of three main points:
1) The authors form the argument that increasing resistance to A.psidii in older leaves is caused by the leaf anatomy changes they presented in their results. However the paper does not include any inoculations with A. psidii to prove that the resistance response would actually occur on the plants they studied in the manner they claim. All the arguments and conclusions about susceptibility/resistance in E. grandis leaves are inferred from other information on the pathogen and disease outside of this paper. This problem could be mitigated and the paper could be made publishable if the authors can adjust the text and language to acknowledge more strongly that their study is only about plant anatomy and not directly about resistance to the pathogen.
However, they also would need to properly mention other possible resistance mechanisms, which they ignore as a result of in their narrow focus on plant anatomy. They need to adequately discuss physiological mechanisms driving resistance development. These operate after the pathogen has infected and may include antifungal compounds, phytoalexins and resistance genes controlling effector proteins and plant biochemical defence pathways. The resistance often observed in Myrtaceae species (including developmental resistance in older leaves) involves a hypersensitive response clearly indicating that post-infection mechanisms of resistance are operating. Statements throughout the paper about the control of resistance through physical barriers need to be made less dogmatic.
For example, L264-266: “Younger plants are more vulnerable to rust because they remain longer during the initial stages (1, 2, and 3) and with a longer maturation period.” The paper provides no direct proof that this statement is true.
Also, the Conclusions (L349-360) are a series of statements that assume anatomical changes are the only mechanism of resistance, when in fact, these could be merely coincidental with the development of physiological resistance mechanisms operating through molecular plant defence pathways.
2) The authors appear confused about the concepts of plant development and plant growth and how these relate to their study. They need to define appropriate and simple terminology for the leaf positions they studied and use it consistently throughout the paper.
At present, the following terms are all used to describe the leaf development stages, when a single term is all that is required:
“host development stages, plant growth stages, phenological stage, phenological leaf-age, stage of maturation, phenological E. grandis leaf stages, leaf phenological stage development, phenological development stages, leaf-age”.
Definitions of plant development, phenology and growth are as follows:
Plant development is a series of identifiable events (growth stages) resulting in a qualitative or quantitative change in plant structure (Dambreville et al. 2015).
Dambreville A, Lauri P, Normand F, Guedon Y 2015. Analysing growth and development of plants jointly using developmental growth stages Annals of Botany 115: 93–105. doi:10.1093/aob/mcu227
These growth stages (often confusingly called phenological stages) are generally divided into vegetative and flowering phases and within each of these is a continuum of organ development that can be further divided into recognisable stages (e.g. BBCH growth stages for crop plants).
Phenology is the study of the time that these events occur and is expressed as the time (date or number of days) or thermal time (degree C days) at which different stages occur. This study does not involve phenology and therefore the words phenology and phenological stages should be dropped. Note that Develey-Rivière & Galiana (2007), who the authors cite, do not use the word phenology in their paper, only development.
Growth, on the other hand, is an irreversible increase in plant or organ dimensions or mass over time within a growth stage (Dambreville et al. 2015). Strictly speaking, the anatomical observations in this study relate to growth and maturity of organs (leaves), not to plant development. The anatomical structures observed (tables 1 and 2) in 6 and 20 month old plants are all present in leaves 1, 3 and 5 and in 6 month and 60 month old plants, they only change in their proportions; there is no qualitative change in the structures that are present.
I suggest referring to the five stages identified in this study only as “developmental stages”, or “growth stages” and do not use the word “phenological” at all.
3) The paper has the materials and methods placed after the results and discussion. I do not know if this is a requirement of the journal, but it detracts from the clarity of the paper. The reader, when reading the results cannot assess them in relation to the methods in order to gauge how the experiments were conducted or how robust they are. The information on variables and factors presented in the tables and figures helps only a little. Before reading the results, the reader needs to know about the methods chosen, the degree of replication and the statistical analyses used. I therefore suggest putting the methods before the results in the conventional manner.
Reviewer 2 Report
The MS reflects nice and accurate work.
The MS achieves the indicated and formulated goals and provides new scientific achievements in the field of "histological mechanism of resistance" research.
Minor errors:
It is recommended to list literary sources according to the numbers in the bibliography.
- ad 100-101 Hickey (1973) & Graf (1970) / (In References: 41 and 42)
- ad 317 (Johansen, 1940)
Species names should be highlighted according to the nomenclature regulations:
- ad 387 (Eucalyptus)
- ad 395 (Corymbia)
- ad 460: (Eucalyptus)
Form errors:
- ad 386: editing error
- Missing space: ad 412, 473, 474
Reviewer 3 Report
The manuscript Phenological stage and morphological characteristics interfering with the resistance of Eucalyptus leaves of two plant ages to Austropuccinia psidii is a well written documents with novel anatomical data that authors hypothetize are valuable to resist E. psidii. There is agreement btween title, objectives and results. However, I consider there are some points that need to be clarify (Table 4) in the results and dicussion. The comments or questions are given in text

Round 2
Reviewer 1 Report
In response to my comments on v1 of this manuscript, I am happy that the authors have adequately addressed point 2), about terminology, and point 3), about position of the materials and methods within the paper.
However, their response does not address point 1), which I confess I may have not made clearly enough in my initial comments on v1. The comment concerns the way presentation of their findings implies that constitutive anatomical changes in leaf anatomy (including essential oil glands) are the only mechanism by which eucalypt leaves change from susceptible to resistant as they grow. They completely overlook the fact that other physiological post-infection resistance mechanisms may also be operating involving a hypersensitive type of host response. This is seen in leaves of some Myrtaceae species as leaf spotting at leaf node positions 4-5 as they transition from susceptible to resistant.
I’m not saying this is necessarily observable in E. grandis, but I am saying that developmental resistance in maturing tissues is a Myrtaceae-wide phenomenon and it can involve physiological rather than anatomical mechanisms and that the authors need to be cognisant of this fact.
L72-73: Regarding reports of age-related (ontogenic) resistance, the authors need to include the previous report and discussion about ontogenic resistance in Myrtaceae by (Beresford et al. 2020):
Beresford RM, Shuey LS, Pegg GS 2020. Symptom development and latent period of Austropuccinia psidii (myrtle rust) in relation to host species, temperature and ontogenic resistance. Plant Pathology 69:484–494. https://doi.org/10.1111/ppa.13145
The authors need to refer to this paper and, in fact, it is a more relevant reference than one given about grapevine [18]. I suggest changing the sentence (L72-73) as follows:
“The effect of plant growth stages on pathogen resistance is known as age-related resistance, developmental resistance or ontogenic resistance and has been previously reported in Myrtaceae [Beresford et al. 2020], as well as in many other plants [19].”
For this study of E. grandis, the anatomical evidence does not rule out the possibility that physiological resistance mechanisms are also operating. The authors need to adjust the text to be less dogmatic about anatomical changes as the causal mechanism of developmental resistance. This is especially the case since the study does not include any direct histopathology that could show how the pathogen interacts with the leaf structures that were observed in the study.
The problem is actually easy for the authors to fix by changing some of their sentences to show awareness that constitutive internal leaf structures may not be the only resistance mechanism operating.
I suggest the following changes:
Abstract L36-37: Change to “Changes in anatomical characteristics that could reduce susceptibility of older E. grandis leaves to A. psidii coincide with the time that leaf resistance develops. Reduced infection of this pathogen on older plants appears to be associated with more rapid maturation of their leaf tissues.”
L368-370: Change to: “Younger plants may be more vulnerable to rust because they remain longer during the initial stages (1, 2, and 3) and have a longer maturation period.
L372-375: Change to: “This phenomenon may reduce the occurrence of the disease due to a shorter exposure period for leaves more susceptible to rust and this could reduce the epidemic period”.
L380-382: Change to: The lower intensity of A. psidii in older plants may be due to a shortened period of high susceptibility associated with more rapid leaf maturation, similar to that reported for the Mycosphaerella sp. and Teratosphaeria sp. on Eucalyptus plants [40].”
L390-393: Change to: “This is the first report demonstrating the natural physical resistance mechanisms of older eucalypt leaves to the Myrtaceae rust, based on morphological and anathomical characteristics, which may explain the lower occurrence of this disease in older eucalypt plants.
L394-396: Change to: The occurrence of reinforced tissues, including sclerenchyma, collenchyma, greater leaf blade thickness, and vascular bundles and the great quantity of cavities containing essential oils are factors that may contribute to older E. grandis leaves being more resistant to rust.
L397-399: Change to: “The absence of lower palisade parenchyma in the and after the fourth leaf stage in young plants and third leaf stage in adult plants could be important in reducing tissue invasion by A. psidii.
L400-402: Change to: “The lower occurrence of the fungus A. psidii in older plants is probably due to the escape of its leaves due to the rapid maturation period (10 days) compared to younger plants under two years old.”
Author Response
We have revised the MS and done all corrections according to the suggestions of the reviewer. We thank the reviewer’s suggestions and comments, which helped us to improve the quality of this text.
We fully agree with its requests that the resistance of eucalypt plants is related to different interactions of chemical and physical resistance mechanisms. This agreement can be seen in the paper's introduction (lines 68-70; 73-75) and our hypothesis (lines 81-82), as well as in the discussion (lines 315-327). However, we include in the article (lines 350-351) that our work provides additional information on the complex mechanism of resistance of eucalypt plants to A. psidii.
